# Impact of double-bolus tracking to individualize scan timing of the portal venous phase in preoperative computed tomography colonography angiography for right-sided colon cancer

Yoshiya Ohashi[1], Masaaki Miyo 🔾[2,3]*, Koichi Okuya[2], Emi Akizuki[2], Atsushi Hamabe[2,4], Ai Noda[2], Masayuki Ishii[2], Ryo Miura[2], Momoko Ichihara[2], Maho Toyota[2], Kohei Okamoto[2], Shun Hayasaka[1], Takeo Tanaka[1], Hiroyuki Takashima[1,5], Kohei Harada 🔾[1], Keishi Ogura[1], Ichiro Takemasa[2,6]*

1 Division of Radiology, and Nuclear Medicine, Sapporo Medical University, Sapporo, Japan, 2 Department of Surgery, Surgical Oncology and Science, Sapporo Medical University, Sapporo, Japan, 3 Department of Gastroenterological Surgery, Osaka International Cancer Institute, Osaka, Japan, 4 Department of Gastroenterological Surgery, Graduate School of Medicine, Osaka University, Suita, Japan, 5 Division of Biomedical Science and Engineering, Faculty of Health Sciences, Hokkaido University, Sapporo, Japan, 6 Department of Surgery, Osaka International Medical and Science Center, Osaka Keisatsu Hospital, Osaka, Japan

* mmiyo@sapmed.ac.jp (MM); itakemasa@sapmed.ac.jp (IT)

## Abstract

### Aim

In computed tomography colonography angiography (CTC-A), used for preoperative screening of right-sided colon cancer, the timing of venous phase imaging is conventionally determined by a fixed-delay time; however, the contrast effect may be insufficient because of individual differences in blood flow status. Therefore, we developed the double-bolus tracking (DBT) method to solve this issue.

### Method

We compared the contrast effect and image quality of the portal venous systems between two methods of the conventional fixed-delay and DBT which utilizes low-dose monitoring to individualize venous scan timings. Data from 30 consecutive patients who underwent CTC-A for right-sided colon cancer using the DBT method were prospectively collected and compared with that from 30 consecutive patients who underwent the conventional fixed-delay method between August 2018 and July 2022. CT values of the portal vein, gastrocolic trunk, and middle colic veins were measured. Additionally, two gastrointestinal surgeons performed a five-point visual evaluation of the three-dimensional volume rendering image of the gastrocolic trunk.

**Data availability statement:** All relevant data are within the manuscript and its Supporting Information files.

**Funding:** The author(s) received no specific funding for this work.

**Competing interests:** The authors have declared that no competing interests exist.

## Results

CT values in the DBT group were significantly higher than those in the fixed-delay group. (portal vein: 266.7 HU vs. 210.0 HU; $p < 0.001$, gastrocolic trunk: 251.6 HU vs. 191.0 HU; $p < 0.001$, middle colic vein: 257.2 HU vs. 190.1 HU; $p < 0.001$). Visual assessment of the gastrocolic trunk was significantly higher in the DBT group than that in the fixed-delay group (DBT, 3.6, 3.4; fixed-delay, 2.6, 2.8; $p = 0.003$, $p = 0.044$).

## Conclusion

The DBT method can enhance the contrast effect of the portal venous systems and improve image quality.

## Introduction

Laparoscopic surgery for colon cancer is equivalent or superior to open surgery in terms of short- and long-term outcomes in multiple large randomized controlled trials and is now broadly applied to reduce the invasiveness of surgeries [1–6]. Robotic surgical systems are promising advanced technologies with several advantages compared to laparoscopic surgeries, with better short-term results than with laparoscopic surgery in rectal cancer, and are also expected to overcome the problems of laparoscopic colectomy, improving the surgical quality of colon cancer [7,8]. Despite the increasing use of minimally invasive surgery, the 90-d postoperative mortality rate for right hemicolectomy is still estimated at approximately 2%, possibly due to many variations in the running of arteries and veins in the right colon [9,10]. Therefore, accurate preoperative evaluation of vessels' anatomical variations is important for safe surgery as well as appropriate lymph node dissection [11–13].

Currently, computed tomography angiography (CTA) can be used to preoperatively identify a patient's vascular structures, and the images can be processed using rendering software to reconstruct a three-dimensional image of the mesenteric vasculature. Such softwares have long been used in vascular surgery and interventional radiology, and CT colonography angiography (CTC-A) has also been used to visualize the anatomical location of arteries, veins, and tumors in the colorectal surgical field [14,15]. Surgical simulation using CTC-A contributes to shorter operating times and lesser blood loss and is useful in improving the patient's quality of life after the laparoscopic surgery [16,17]. The image quality of the appropriate CTA is related to the injection method of the contrast agent and timing of the imaging [18–22]. In aortic angiography, auto-bolus tracking technology, which individually modulates the timing of arterial phase imaging, is useful in reducing variability in contrast effect and improving diagnostic performance [23,24]. In contrast, CT portal venography is generally performed with a fixed-delay time between the arterial and venous phases; however, achieving contrast enhancement is sometimes difficult because the contrast agent perfusing visceral organs and tissues mixes with the blood and dilutes it [25,26]. Previous studies have reported techniques for imaging the portal venous phase using low-dose monitoring [27–29]. However, the surgeon also needs to confirm the variations of the arteries before surgery for right-sided colon cancer, and simply optimizing the portal venous phase is not enough information for a preoperative CTC-A. Therefore, we developed a double-bolus tracking (DBT) method that allows low-dose monitoring to be used twice in one imaging series. We aimed to clarify the effect of the DBT method on the contrast enhancement and image quality of the venous phase in CTC-A compared to the fixed-delay method.

## Materials and methods

### Clinical information

This study was approved by the Institutional Review Board of Sapporo Medical University (approval number 312-180) and registered in the UMIN Clinical Trials Registry (UMIN000043417). All patients in the DBT group provided written informed consent after the nature of the procedure was fully explained. Patients in the fixed-delay group could opt out of the study via the hospital website. The patients' electronic medical records were accessed for research purposes from 16/01/2020 to 31/03/2024.

We prospectively collected data from 30 consecutive patients who underwent CTC-A with the DBT method for right-sided colon cancer at our institution between January 2020 and July 2022 and compared the data with the retrospectively collected data of 30 consecutive patients who underwent CTC-A using the conventional fixed-delay method from August 2018 to November 2019. Exclusion criteria were as follows: previous surgery for gastrointestinal disease, if the treating physician considered enrolment in this study inappropriate, history of severe side effects of contrast media, or if the patient could not consent to participate in the study.

All patients underwent standard colonic irrigation and endoscopy, followed by CTC-A. The patients were asked to take 24 mg of Sennoside A·B Calcium (Sennoside Tablets, Nichi-Iko Pharmaceutical Co., Ltd.) on the day before and morning of the examination. Additionally, 2,000 mL of polyethylene glycol (Moviprep, EA Pharma Co., Ltd.) was administered for bowel cleansing and 1 mg of glucagon (Glucagon G Novo Injection, Novo Nordisk Pharma Ltd.) or 20 mg of Scopolamine Butylbromide (Buscopan Injection, Sanofi K.K.) was administered via intramuscular injection just before the endoscopy. If no complications were noted, CTC-A was administered immediately after the colonoscopy. An enema tube was inserted into the anus before the CT scan, and a total of 2.0–3.0 L (20 mmHg gas injection pressure) of carbon dioxide gas was used to dilate the colon using automated carbon dioxide insufflation (KSC-130, Kyorin Systemac).

### Scanning techniques

Multidetector CT was performed using an 80-row CT system (Aquilion Prime, Canon Medical Systems). Acquisition parameters were as follows: detector collimation, $0.5 \times 80$ mm; pitch factor, 0.813–1.388; tube voltage, 120 kVp; tube current auto-modulation (noise index 8 at 5.0 mm slice thickness); rotation time, 0.5 s; scan field of view, 280–380 mm. A 20 or 22-gauge plastic intravenous catheter was placed into the antecubital vein. Nonionic contrast agent (iopamidol 370 and iopamilon 370, Bayer; or Iohexol [Omnipaque 300, GE Healthcare Pharma]; or iomeprol [Iomeron 350, Eisai]) was injected using a power injector (Dual shot GX 7, Nemoto Kyorindo). A total contrast dose of 600 mgI/kg was rapidly injected at a rate of 3.6–5.4 mL/s over 25 seconds, followed by a 40 mL flush of 0.9% saline at the same flow rate [22]. A bolus-tracking system (Real Prep, Canon Medical Systems) was used to determine the start time of the scan for the arterial phase, which was used to construct arterial 3D-VR reconstructions in both the DBT and fixed-delay methods. Low-dose monitoring scans (detector collimation, $0.5 \times 4$ mm; tube voltage, 120 kV; tube current, 50 mA) were started continuously at the position of the diaphragmatic dome, and the arterial phase was acquired when the abdominal aortic region of interest (ROI) reached a threshold of 250 Hounsfield units (HU). In the DBT group, a low-dose monitoring scan (beam collimation, $0.5 \times 40$ mm; tube voltage, 120 kV; tube current, 50 mA) was initiated at 1.5-s intervals at the level of the hepatic portal region after the arterial phase, and the portal venous phase was initiated 5 s after reaching the threshold of 250 HU enhancement in the portal vein. The volume CT dose index ($CTDI_{vol}$), dose-length product (DLP), and

number of scans for low-dose monitoring were obtained from automatically generated radiation dose reports. In the fixed-delay group, the portal venous phase was acquired with a delay time of 25 s after the completion of the arterial phase acquisition (Fig 1).

All images were reconstructed in three planes (axial, coronal, and sagittal) with a slice thickness of 2.0 mm using an abdominal kernel and adaptive iterative dose reduction (AIDR 3D enhanced standard, Canon Medical Systems). Additionally, a three-dimensional volume rendering (3D-VR) was reconstructed using a workstation (ZIOSTATION2, AMIN Corporation).

### Image analysis

Quantitative measurements were performed by a blinded reader (S.H., 9 years of experience). The ROI size was approximately 10 mm$^2$, and the CT values were obtained from the portal vein (PV) at the hepatic hilum, the root of the gastrocolic trunk (GCT) draining into the superior mesenteric vein (SMV), the root of the inferior mesenteric vein (IMV) draining into the splenic vein (SPV) or the SMV, and the root of the middle colic vein (MCV) draining into the SMV (Fig 2). Additionally, an ROI of approximately 5 cm$^2$ was placed in the erector spinae at the umbilical level and the mean CT values were measured (ROI $_{muscle}$). Based on these measurements, the contrast-to-noise ratio (CNR) of the vessels was calculated using the following equations [30].

$$CNR_{ROI} = (ROI_{\ standard} - ROI_{\ muscle})/SD_{\ fat}$$

where ROI $_{muscle}$ is the mean attenuation of the erector spinae muscles and SD $_{fat}$ is the standard deviation of pixel values in uniform abdominal wall fat on the anterior abdominal wall. Two

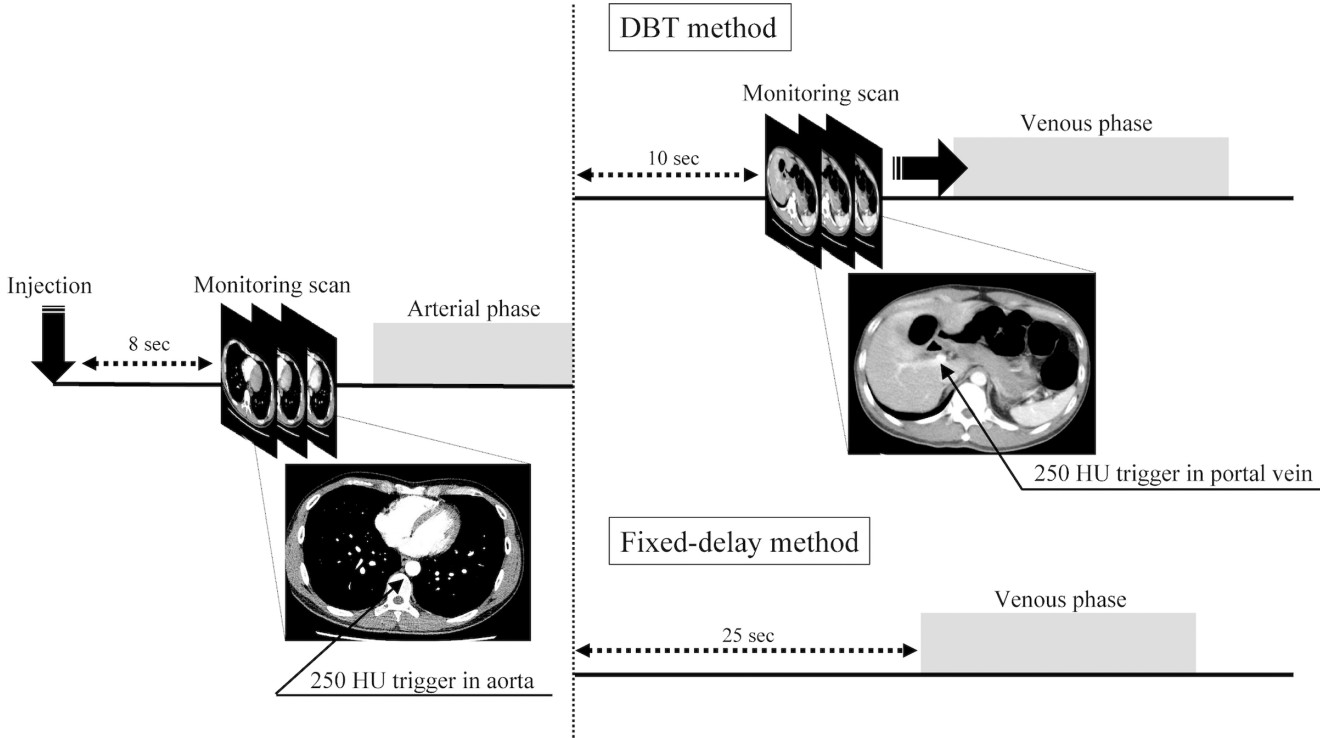

**Fig 1. Scanning protocol for DBT and fixed-delay methods.** DBT, double-bolus tracking.

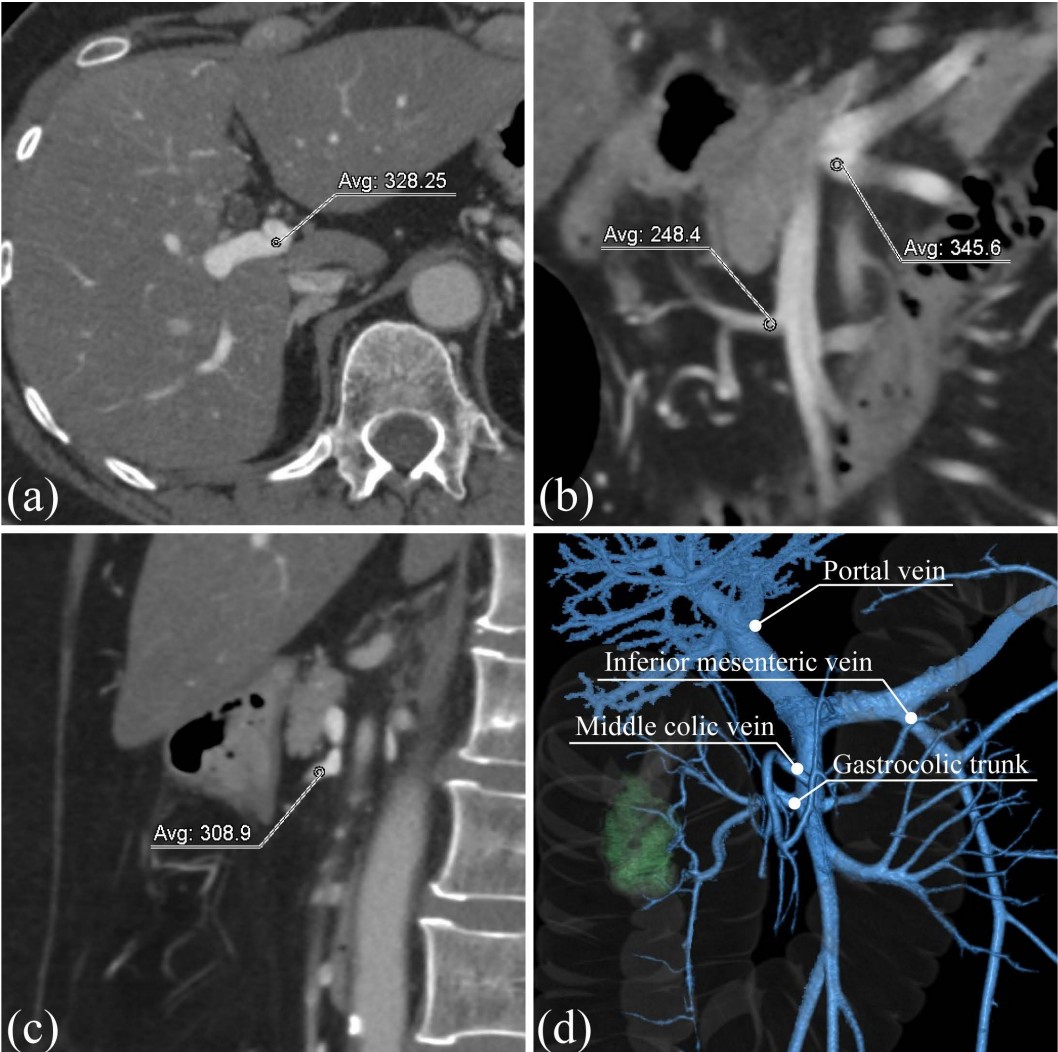

**Fig 2. Venous vascular measuring point.** (a) The portal vein at the hepatic hilum. (b) The root of the gastrocolic trunk (GCT) draining into the superior mesenteric vein (SMV). The root of the inferior mesenteric vein (IMV) draining into the splenic vein (SPV) or the SMV. (c) The root of the middle colic vein (MCV) draining into the SMV. (d) Three-dimensional volume rendering (3D-VR).

blinded surgeons (M.M. and K.O.; 17 and 8 years of experience, respectively) scored the image quality of the reconstructed 3D-VR using a 5-point Likert scale [31]. The 3D-VR threshold was set as the default display setting. Two gastrointestinal surgeons individually evaluated image quality as follows: 1) nondiagnostic (no GCT delineated), 2) poor (only the base of the GCT delineated), 3) acceptable (the distal portion of the GCT delineated), 4) good (terminal branches of the GCT delineated), and 5) excellent (terminal branches of the GCT clearly delineated).

## Statistical analysis

Statistical analyses were performed using the JMP Pro 17 software (SAS Institute Inc., Cary, NC, USA). $\chi^2$ tests were used to test for sex and tumor location differences between the two groups. The Student t-test was used to compare patient characteristics and quantitative and

qualitative data between the two groups. *P*-values were two-tailed, with < 0.05, defined as indicating a statistically significant difference. For qualitative analysis, interobserver agreement was calculated using Cohen's weighted kappa statistic to assess the agreement between the two readers. A kappa value of ≤ 0.20 was interpreted as slight agreement, 0.21–0.40 as fair agreement, 0.41–0.60 as moderate agreement, 0.61–0.80 as substantial agreement, and ≥ 0.81 as almost perfect agreement [32].

## Results

### Patient characteristics

A total of 60 patients (30 in the DBT group and 30 in the fixed-delay group) were included. Patient characteristics, such as age, sex, body mass index, and tumor location, were not significantly different between the two groups (Table 1). The median $CTDI_{vol}$ for low-dose monitoring in the DBT group was 8.1 mGy (range, 2.7–16.2) and the median DLP was 16.2 mGy-cm (range, 5.4–32.4). The median number of low-dose monitoring scans was 3 (range, 1–6), and the median delay from the end of the arterial phase to the start of the portal phase in the DBT group was 23.4 s (range, 20.4–27.6) (95% confidence interval 0.3–1.9; $p = 0.005$), compared to 25 s in the fixed-delay group. Eighteen of 30 patients (60.0%) in the DBT group were scanned before the 25-s delay time applied in the fixed-delay group, and two of these 18 patients were started early, immediately after monitoring, while 12 of 30 (40.0%) patients were scanned with a delay time of 25–28 s (Fig 3a).

### Quantitative analysis

The CT values of the veins and the CNR results are shown (Figs 3b and 3c). The mean CT values ± standard error of all measured veins in the DBT group were significantly higher than those of the fixed-delay group: PV (DBT, 266.7 ± 6.7 HU; fixed-delay, 210.0 ± 4.7 HU; $p < 0.001$), GCT (DBT, 251.6 ± 7.4 HU; fixed-delay, 191.0 ± 4.9 HU; $p < 0.001$), MCV (DBT, 257.2 ± 8.6 HU; fixed-delay, 190.1 ± 5.9 HU; $p < 0.001$), IMV (DBT, 282.7 ± 10.9 HU; fixed-delay, 216.6 ± 6.2 HU; $p < 0.001$). Similar to the CT values, venous CNR ± standard error was also significantly higher in the DBT group than that in

the fixed-delay group, with particularly large differences in IMV: PV (DBT, 33.1 ± 1.8; fixed-delay, 20.0 ± 1.3; $p < 0.001$), GCT (DBT, 30.4 ± 1.7; fixed-delay, 17.5 ± 1.1; $p < 0.001$), MCV (DBT, 31.5 ± 1.9; fixed-delay, 17.4 ± 1.2; $p < 0.001$), and IMV (DBT, 35.8 ± 2.4; fixed-delay, 20.7 ± 1.2; $p < 0.001$) (Table 2). With regard to the accessory right colic vein (ARCV), its identification was difficult in 13.3% of cases (4/30) in the fixed-delay group

**Table 1. Patient characteristics.**

| Characteristic | DBT group (N = 30) | Fixed-delay group (N = 30) | *p*-value |
|---|---|---|---|
| Age, years ※ | 70.7 ± 2.2 | 66.5 ± 1.8 | 0.085 |
| Sex | | | 0.295 |
| Male/Female | 10/20 | 14/16 | |
| Body mass index, kg/m²※ | 23.2 ± 0.8 | 22.7 ± 0.7 | 0.344 |
| Tumor location | | | 0.685 |
| Cecum | 5 | 4 | |
| Ascending | 18 | 16 | |
| Transverse | 7 | 10 | |

※ Data are mean ± standard error.

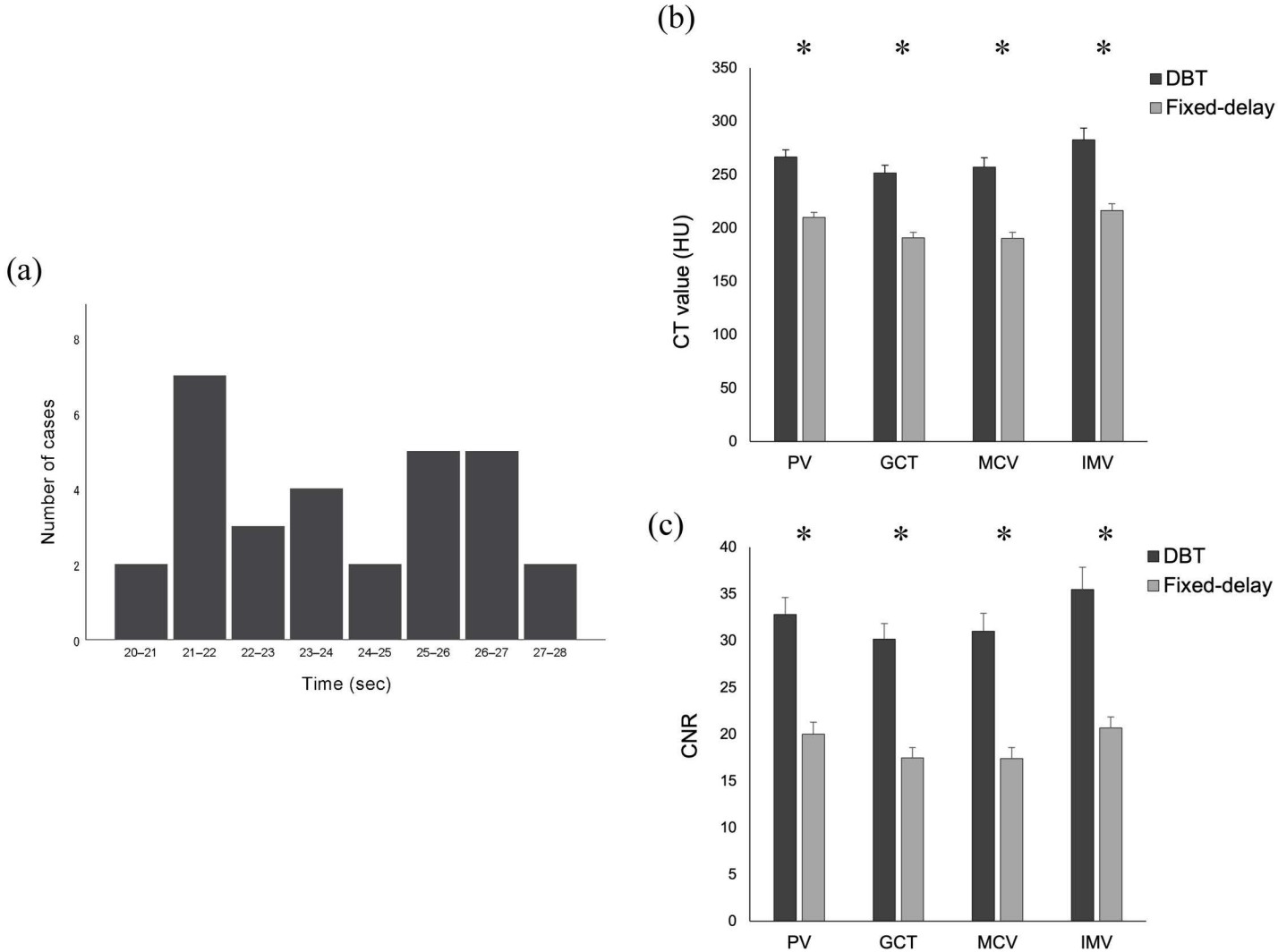

**Fig 3. Venous phase scan timing, computed tomography (CT) values, and the contrast-to-noise ratio (CNR) in the double-bolus tracking (DBT) method compared with the conventional fixed-delay method.** (a) Cumulative histogram showing the trigger delay in the DBT method (range, 20.4–27.6 s; median, 23.4 s). (b) (c) Bar graph showing the comparison of CT value and CNR for the portal, gastrocolic, middle colic, and inferior mesenteric veins between the DBT and fixed-delay groups (mean ± standard error, * $p < 0.001$).

compared to 3.3% (1/30) in the DBT group. Excluding these cases, its mean CT values were 144.2 ± 9.3 HU in the DBT group and 135.1 ± 6.7 HU in the fixed-delay group, with no significant difference ($p = 0.447$). (S1 Table).

## Qualitative analysis

The results of the two gastrointestinal surgeons' individual five-point ratings of the 3D-VR image quality are shown (Fig 4). The image quality ratings of 3D-VR with DBT were rated good or excellent in 20 (66.7%) and 17 of 30 (56.7%) patients, respectively, by the two gastrointestinal surgeons, while those with fixed-delay were good or excellent in 9 of 30 (30.0%) patients for both the surgeons. The mean scores for 3D-VR image quality in the DBT group were significantly higher for both surgeons compared with those in the fixed-delay group: mean scores ± standard error for each of the two surgeons (DBT, 3.6 ± 0.2, 3.4 ± 0.3;

**Table 2. Differences in the CT values and contrast-to-noise ratios.**

|  | DBT group (N = 30) | Fixed-delay group (N = 30) | Difference | p-value |
|---|---|---|---|---|
| CT value (HU) ※ |  |  |  |  |
| PV | 266.7 ± 6.7 | 210.0 ± 4.7 | 56.7 | < 0.001 |
| GCT | 251.6 ± 7.4 | 191.0 ± 4.9 | 60.6 | < 0.001 |
| MCV | 257.2 ± 8.6 | 190.1 ± 5.9 | 67.1 | < 0.001 |
| IMV | 282.7 ± 10.9 | 216.6 ± 6.2 | 66.1 | < 0.001 |
| Contrast-to-noise ratio※ |  |  |  |  |
| PV | 33.1 ± 1.8 | 20.0 ± 1.3 | 13.1 | < 0.001 |
| GCT | 30.4 ± 1.7 | 17.5 ± 1.1 | 12.9 | < 0.001 |
| MCV | 31.5 ± 1.9 | 17.4 ± 1.2 | 14.1 | < 0.001 |
| IMV | 35.8 ± 2.4 | 20.7 ± 1.2 | 15.1 | < 0.001 |

※ Data are presented as mean ± standard error.

CT, computed tomography; HU, Hounsfield unit; DBT, double-bolus tracking; PV, portal vein; GCT, gastrocolic trunk; MCV, middle colic vein; IMV, inferior mesenteric vein.

fixed-delay, 2.6 ± 0.2, 2.8 ± 0.2; $p = 0.003$, $p = 0.044$). The kappa value showed fair and slight agreement between two readers (kappa value = 0.45, 0.39).

## CTC-A in clinical practice

Figs 5a and 5b indicate a 3D-VR of a man with transverse colon cancer, imaged using the DBT and fixed-delay methods. Both the DBT and fixed-delay methods were performed on the same patient in only one of the 60 cases, and the imaging conditions and contrast volume were the same except for the scan timing. The ARCV and the right gastroepiploic vein (RGEV) were more clearly delineated in 3D-VR using the DBT method than that using the fixed-delay method. Fused 3D-VR images of arteries, veins, and pancreas are shown (Fig 5c). There were no significant differences in clinical outcomes, including conversion rate to open surgery (3.6% vs 0%), operative time (312.2 ± 20.4 minutes vs 361.0 ± 25.7 minutes), blood loss (40.1 ± 27.5 mL vs 51.8 ± 33.4 mL), postoperative complications (7.1% vs 5.3%), and length of hospital stay (8.4 ± 1.2 days vs 7.9 ± 1.5 days), between the DBT and fixed-delay groups.

## Discussion

The conventional fixed-delay method, which is commonly used as the imaging method for CTC-A, has a fixed-delay time, which makes optimizing the timing of vein imaging difficult because of the inability to accommodate differences in blood circulation and physiological variations between patients. Consequently, the 3D-VR images used during preoperative simulation are unclear, which may have resulted in difficulty for surgeons in understanding the variation of vein runs [33]. Our analysis showed a range of delay times in the DBT group from 20.4–27.6 s compared to the standard 25-s protocol, suggesting that there are individual differences in contrast peak timing of portal veins. In a previous study, scans while monitoring attenuation of the hepatic parenchyma using a 50 HU threshold significantly increased enhancement of the hepatic parenchyma, portal vein, and hepatic veins when compared to a fixed-delay protocol, suggesting the importance of capturing appropriate contrast timing in individual patients as in this study [34]. The timing of contrast peaks in organs has been suggested to be related to a variety of factors, including body weight, metabolic status, cardiac output, degree of hydration, and development of liver disease [35–37]. Therefore, as proper scan timing in the venous phase is difficult to achieve using only the bolus tracking technique

(a)

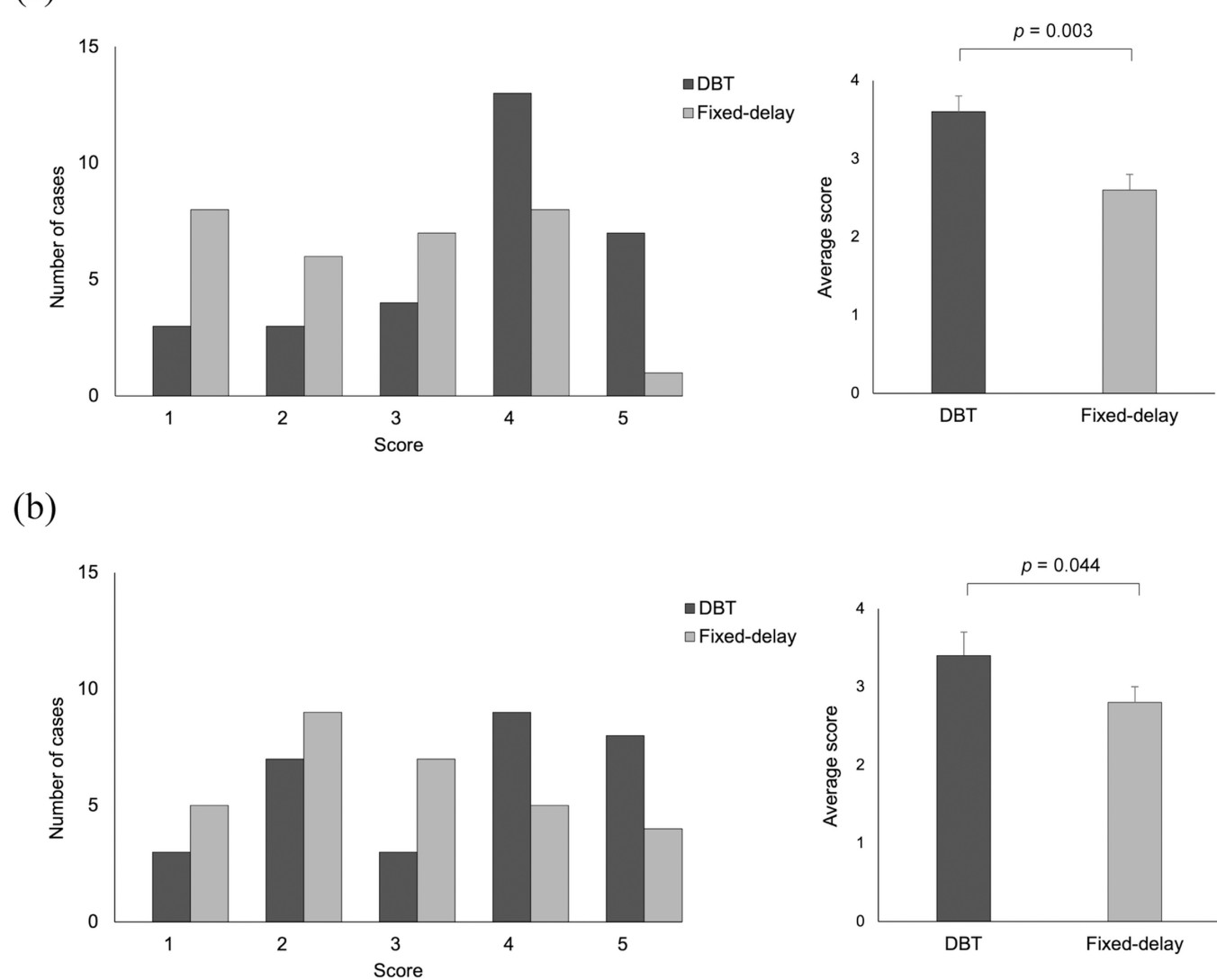

(b)

**Fig 4. Image quality scoring of three-dimensional volume rendering (3D-VR) images evaluated using a five-point scale by two blinded surgeons.** Image quality is scaled as follows: 1) nondiagnostic (no GCT delineated), 2) poor (only the base of the GCT delineated), 3) acceptable (the distal portion of the GCT delineated), 4) good (terminal branches of the GCT delineated), and 5) excellent (terminal branches of the GCT clearly delineated). a and b represent the results of each surgeon. Comparison of the number of each score and the average of those scores (mean ± standard error).

utilized in the arterial phase, a new imaging technique called the DBT method was developed as a solution. The DBT technique allows real-time monitoring of the contrast agent's dynamics, enabling enhancement of the contrast effect without increasing the amount of contrast agent, by accurately determining the optimal timing for its administration. Furthermore, it can be widely implemented regardless of the imaging area or equipment because it does not require specialized applications for hemodynamic monitoring or complex test bolus methods involving small amounts of contrast agents. During low-dose monitoring, precisely identifying the target blood vessels within a short timeframe is crucial. However, the commonly chosen SMV is not easily identified as a monitoring vessel owing to its thinness and location, and may not be suitable as a target vessel. Our DBT method selects the relatively larger-diameter

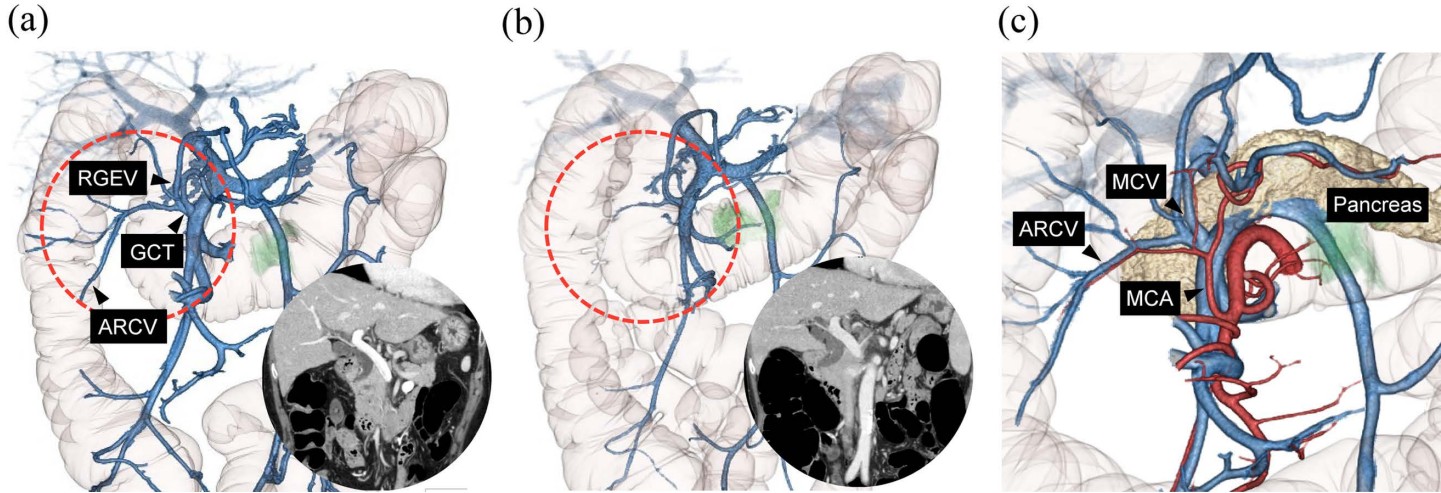

**Fig 5. Representative three-dimensional volume rendering (3D-VR) images of transverse colon cancer.** (a) The 3D-VR images obtained using the double-bolus tracking (DBT) method clearly show the gastrocolic trunk (GCT) consisting of the accessory right colic vein (ARCV) and right gastroepiploic vein (RGEV). (b) The 3D-VR images obtained using the conventional fixed-delay method do not clearly visualize the GCT. (c) Fused image of arteries and veins without mis-registration used in surgical simulation. The relationship between the middle colic artery (MCA), middle colic vein (MCV), accessory right colic vein (ARCV), and pancreas is clear.

hepatic portal vein where the SMV and SPV converge and provides easier visibility during monitoring, reducing the likelihood of interobserver errors.

From the viewpoint of reducing exposure, the blood flow velocity in veins is relatively slow compared to that in arteries, and even if the monitoring scan is not continuous, such as an intermittent scan, it will not affect the capture of the appropriate scan timing if the imaging technologist can estimate the contrast peak timing of the portal vein. However, in the DBT group, 2 out of 30 patients (6.7%) had an extremely fast venous circulation velocity and achieved the portal threshold at the start of monitoring, making it difficult to capture the appropriate start time by manual operation. This may be a limitation of the DBT. One potential solution to address this issue is to start monitoring earlier, although this might increase the risk of radiation exposure associated with the monitoring process. Although it has been reported that inter-patient variations in the contrast peak timing of the portal venous system are related to physiological and physical factors such as cardiac function or liver condition, evidence is insufficient to provide a clear rationale for these connections. Further studies will reveal the crucial factors in the contrast peak timing of the portal venous system, which will lead to an estimation of the scan timing of the portal venous system and a further reduction in monitoring-related radiation exposure. The greatest advantage of the DBT method is that it clarifies the 3D-VR of arteries and veins necessary for surgical simulations (Fig 5c). However, when overlaying 3D-VR fused at different phases, there is a concern regarding the displacement of arterial and venous positional relationships based on the patient's respiratory stopping position and organ movements. According to our previous study, leveraging nonrigid registration techniques through nonlinear image processing programs can provide accurate anatomical information without positional discrepancies [38].

The significance of prophylactic measures against bleeding is widely recognized, as the frequency of vascular complications of intraoperative bleeding in patients with colon cancer undergoing laparoscopic right hemicolectomy was reported to be 5.4%, of which 3.3% were from the GCT, pancreatic head, and ileocolic artery [39]. GCT, observed in 86% of cases, shows high variability, with the most frequent variation in which RGEV, anterior superior pancreaticoduodenal vein (ASPDV), and ARCV merge into the GCT (42.5%), followed by

that in which RGEV, ASPDV, ARCV, and right colic vein merge into the GCT (20.1%) [40]. Rare cases with vascular running variation have been noted where the GCT and ileocolic vein (ICV) form a common trunk with variant courses, and because the convergence points of ICV and SMV is one of the landmarks for lymph node dissection, understanding these variations of GCT through 3D-VR is clinically important for ensuring a safe lymph node dissection. Moreover, the drainage pathway of the MCV is classified into five types based on a frequency pattern of SMV in 62.5%, GCT in 29.3%, IMV in 4.8%, SPV in 2.7%, and first jejunal vein in 0.6%, which is the most complex variation of the vessels to recognize during right hemicolectomy and requires attention [41–43]. In previous studies, intraoperative ultrasound (IOUS) has been reported as a method of understanding the location of SMV; however, IOUS has not been widely adopted in actual clinical practice due to the experience required of the ultrasound operator [44]. The advantage of the DBT method is to create clear 3D-VR reconstructions from the well-enhanced CTC-A, allowing objective surgical planning without the need for advanced technical skills. DBT may prevent complications such as intraoperative bleeding by visually clarifying the positional relationship of venous branches, which may lead to improved quality of life for patients in the future. While initial costs and workflow adjustments, such as staff training and equipment upgrades, are required, the DBT method's potential to improve diagnostic accuracy justifies these investments in well-equipped facilities, which could enhance its generalizability in the future.

This study had some limitations. First, the DBT method may vary because the threshold for contrast enhancement was determined manually rather than automatically. Second, further validation in future studies is necessary due to the small number of patients included in this study. The slight increase in radiation exposure due to additional low-dose scans is another limitation. Although minimized through optimized protocols, this remains a concern, especially for patients requiring repeated imaging. Future improvements in low-dose techniques are essential to address this issue and enhance patient safety.

## Conclusions

Contrast enhancement and image quality of preoperative CTC-A for right-sided colon cancer may be improved by the DBT method, which optimizes the venous phase scan timing.

## Supporting information

**S1 Table. CT values and contrast-to-noise ratios for accessory right colic vein.**
(DOCX)

## Acknowledgments

We would like to thank Editage (www.editage.jp) for English language editing.

## Author contributions

**Conceptualization:** Yoshiya Ohashi, Masaaki Miyo, Atsushi Hamabe, Ichiro Takemasa.

**Data curation:** Yoshiya Ohashi, Masaaki Miyo, Koichi Okuya, Ai Noda, Masayuki Ishii, Ryo Miura, Momoko Ichihara, Maho Toyota, Kohei Okamoto, Shun Hayasaka, Takeo Tanaka.

**Formal analysis:** Yoshiya Ohashi.

**Funding acquisition:** Koichi Okuya, Kohei Harada, Ichiro Takemasa.

**Investigation:** Yoshiya Ohashi, Masaaki Miyo, Koichi Okuya, Emi Akizuki, Kohei Okamoto, Shun Hayasaka.

**Project administration:** Yoshiya Ohashi, Masaaki Miyo, Atsushi Hamabe, Kohei Harada, Keishi Ogura, Ichiro Takemasa.

**Resources:** Masaaki Miyo, Koichi Okuya, Atsushi Hamabe, Masayuki Ishii.

**Supervision:** Masaaki Miyo, Koichi Okuya, Ichiro Takemasa.

**Visualization:** Yoshiya Ohashi, Hiroyuki Takashima, Ichiro Takemasa.

**Writing – original draft:** Yoshiya Ohashi, Masaaki Miyo.

**Writing – review & editing:** Masaaki Miyo, Koichi Okuya, Ichiro Takemasa.

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
