## [Decision Letter · Decision Letter 0]

23 Jan 2025

PONE-D-24-39564Impact of double-bolus tracking to individualize scan timing of the portal venous phase in preoperative computed tomography colonography angiography for right-sided colon cancerPLOS ONE

Dear Dr. Miyo,

Thank you for submitting your manuscript to PLOS ONE. After careful consideration, we feel that it has merit but does not fully meet PLOS ONE’s publication criteria as it currently stands. Therefore, we invite you to submit a revised version of the manuscript that addresses the points raised during the review process.

We look forward to receiving your revised manuscript.

Kind regards,

Tsutomu Kumamoto

Academic Editor

PLOS ONE

Journal Requirements:

2. Please remove all personal information, ensure that the data shared are in accordance with participant consent, and re-upload a fully anonymized data set. 

Additional Editor Comments :

Thank you for submitting your manuscript titled "Impact of double-bolus tracking to individualize scan timing of the portal venous phase in preoperative computed tomography colonography angiography for right-sided colon cancer."

I found your study to be both innovative and clinically relevant. However, I would like to provide some comments and suggestions for further improving the manuscript:

1. Evaluation of ARCV

I suggest including the accessory right colic vein (ARCV) as part of the venous evaluations performed using the DBT method. The ARCV is a crucial anatomical structure in right-sided colon cancer surgeries, and its clear delineation during preoperative planning may help reduce the risk of surgical complications, such as unplanned conversions to open surgery.

2. 3D Reconstruction of Arterial Structures

Please clarify whether arterial 3D-VR reconstructions were performed using the DBT method and if there were any notable differences compared to the conventional method.

3. Comparison of Clinical Outcomes

It would be valuable to assess whether the DBT method influenced clinical outcomes compared to the conventional method. For example, were there any differences in conversion rates to open surgery, operative time, blood loss, complications, or length of hospital stay? Including data on these short-term outcomes could provide more practical insights into the clinical impact of your findings.

4. Generalizability, Cost, and Patient Burden

Please include a discussion about the feasibility of implementing the DBT method in general facilities, considering workflow changes, costs, and required resources. Additionally, mention potential patient burden, such as increased radiation exposure due to low-dose monitoring, as part of the "Limitations" section, which could enhance the manuscript's practical relevance.

Reviewers' comments:

Reviewer's Responses to Questions

**Comments to the Author**

1. Is the manuscript technically sound, and do the data support the conclusions?

Reviewer #1: Yes

2. Has the statistical analysis been performed appropriately and rigorously?

Reviewer #1: Yes

3. Have the authors made all data underlying the findings in their manuscript fully available?

Reviewer #1: Yes

4. Is the manuscript presented in an intelligible fashion and written in standard English?

Reviewer #1: Yes

5. Review Comments to the Author

Reviewer #1: This is a well written article covering an interesting topic: the use of bolus tracking techniques for improving venous phase imaging.

I hove some suggestions for further improving your manuscript:

- Abstract: in the "aim" section there is some information that should be moved in the "method" one

- Page 6 line 81: another recently published article evaluated the use of bolus tracking technique for PVP CT (doi: 10.1007/s00330-024-11009-7). I suggest you to add it as a reference.

- Page 7, lines 97-103: move this information to the beginning of the subheading

- Page 97, lines 104-108: please better explain the preparation with dosages

- Page 8, lines 121-123: please clarify ho CM was dosed and which injection flow was used

- Page 9, lines 128-129: was there any delay between aortic peak and scanning start?

6. PLOS authors have the option to publish the peer review history of their article (what does this mean? ). If published, this will include your full peer review and any attached files.

**Do you want your identity to be public for this peer review?** For information about this choice, including consent withdrawal, please see our Privacy Policy .

Reviewer #1: **Yes: ** Matteo Bonatti

---

## [Author Response · Author response to Decision Letter 1]

20 Feb 2025

We thank the Editor and reviewers for the fair comments and useful suggestions to improve our manuscript. As indicated below, we have taken the comments and suggestions into consideration, and we made corrections on a proof in red for the revised part of the manuscript. And the number of decimal places for the p-values has been unified throughout.

Additional Editor Comments:

1. Evaluation of ARCV

I suggest including the accessory right colic vein (ARCV) as part of the venous evaluations performed using the DBT method. The ARCV is a crucial anatomical structure in right-sided colon cancer surgeries, and its clear delineation during preoperative planning may help reduce the risk of surgical complications, such as unplanned conversions to open surgery.

Thank you for your helpful suggestion. We evaluated the enhancement of the accessory right colic vein (ARCV) by measuring its CT values. Identification of the ARCV was difficult in 13.3% of cases (4/30) in the fixed-delay group compared to 3.3% (1/30) in the DBT group, and excluding these cases, the mean CT values were 144.2 ± 9.3 HU in the DBT group and 135.1 ± 6.7 HU in the fixed-delay group. Both imaging performance and enhancement of ARCV tended to be better with the DBT method compared to the fixed delay method, but the difference was not significant. These results might be due in part to the small sample size in our study. We described these findings on page 15 lines 219-223 of the “Results” section and added S1 Table about these as supporting information.

2. 3D Reconstruction of Arterial Structures

Please clarify whether arterial 3D-VR reconstructions were performed using the DBT method and if there were any notable differences compared to the conventional method.

In the DBT method, 3D volume-rendered (VR) images of the arteries were also reconstructed in as well as the fixed-delay method. Since both the DBT and conventional methods rely on the same arterial enhancement phase, there is no substantial difference in arterial visualization between the two approaches. We have modified the description about this to make it clearer on page 9 lines 129~130 of “Materials and methods” section.

3. Comparison of Clinical Outcomes

It would be valuable to assess whether the DBT method influenced clinical outcomes compared to the conventional method. For example, were there any differences in conversion rates to open surgery, operative time, blood loss, complications, or length of hospital stay? Including data on these short-term outcomes could provide more practical insights into the clinical impact of your findings.

We appreciate your insightful comments. The mean operative time was 312.2 ± 20.4 minutes in the DBT group and 361.0 ± 25.7 minutes in the fixed-delay group (p = 0.135), blood loss was 40.1 ± 27.5 mL and 51.8 ± 33.4 mL (p = 0.787), and conversion to open surgery was required in 3.6% and 0% of cases (p = 1.000). Postoperative complications occurred in 7.1% of cases in the DBT group and 5.3% in the fixed-delay group (p = 1.000), and the mean length of stay was 8.4 ± 1.2 days and 7.9 ± 1.5 days (p = 0.780), with no significant differences between the groups. Limitations such as case selection bias and small sample size should be considered when discussing these clinical outcomes. We added the comments about these on page 18 lines 266-271 of the “Results” section.

4. Generalizability, Cost, and Patient Burden

Please include a discussion about the feasibility of implementing the DBT method in general facilities, considering workflow changes, costs, and required resources. Additionally, mention potential patient burden, such as increased radiation exposure due to low-dose monitoring, as part of the "Limitations" section, which could enhance the manuscript's practical relevance.

We appreciate your helpful advice. While initial costs and workflow adjustments, such as staff training and equipment upgrades, are required, the DBT method’s potential to improve diagnostic accuracy justifies these investments in well-equipped facilities, which could enhance its generalizability in the future. The slight increase in radiation exposure due to additional low-dose scans is another limitation. Although optimized protocols minimized radiation exposure, this remains a concern, especially for patients requiring repeated imaging. Future improvements in low-dose techniques are essential to address this issue and enhance patient safety. We added the comments about these in lines 356-359 and lines 363-367 of the “Discussion” section.

Reviewer #1: This is a well written article covering an interesting topic: the use of bolus tracking techniques for improving venous phase imaging.

I hove some suggestions for further improving your manuscript:

- Abstract: in the "aim" section there is some information that should be moved in the "method" one

Thank you for your helpful comments. We have revised the abstract by moving the information from the “Aim” section to lines 30-32 of the “Method” section.

- Page 6 line 81: another recently published article evaluated the use of bolus tracking technique for PVP CT (doi: 10.1007/s00330-024-11009-7). I suggest you to add it as a reference.

We appreciate your suggestion. We added the recommended paper as a reference on page 6 line 81 of the “Introduction” section.

- Page 7, lines 97-103: move this information to the beginning of the subheading

According to your advice, we moved this information to lines 91-96 in the “Materials and methods” section.

- Page 97, lines 104-108: please better explain the preparation with dosages

According to your advice, the dosage and administration of drugs for preparation were specified as 24 mg of Sennoside A‧B Calcium (Sennoside Tablets, Nichi-Iko Pharmaceutical Co., Ltd.), 1 mg of glucagon (Glucagon G Novo Injection, Novo Nordisk Pharma Ltd.), and 20 mg of Scopolamine Butylbromide (Buscopan Injection, Sanofi K.K.) on page 7-8 lines 106-111 in the “Materials and Methods” section.

- Page 8, lines 121-123: please clarify how CM was dosed and which injection flow was used

We appreciate your valuable comments. The contrast injection protocol was based on the study by Ichikawa T, et al, in which a total dose of 600 mgI/kg injected at 3.6-5.4 mL/s over 25 seconds (1). These details have been included on page 8-9 lines 123-127 of the “Materials and methods” section.

- Page 9, lines 128-129: was there any delay between aortic peak and scanning start?

In our study, the arterial phase imaging method was the same between the DBT and fixed-delay methods, with no significant delay observed in both the DBT group (30 patients) and the fixed-delay group (30 patients).

References

1. Ichikawa T, Erturk SM, Araki T. Multiphasic contrast-enhanced multidetector-row CT of liver: contrast-enhancement theory and practical scan protocol with a combination of fixed injection duration and patients' body-weight-tailored dose of contrast material. Eur J Radiol. 2006;58(2):165-76.

---

## [Editor Report · Decision Letter 1]

23 Feb 2025

Impact of double-bolus tracking to individualize scan timing of the portal venous phase in preoperative computed tomography colonography angiography for right-sided colon cancer

PONE-D-24-39564R1

Dear Dr. Masaaki Miyo,

We’re pleased to inform you that your manuscript has been judged scientifically suitable for publication and will be formally accepted for publication once it meets all outstanding technical requirements.

Kind regards,

Tsutomu Kumamoto

Academic Editor

PLOS ONE

Additional Editor Comments:

Congratulations on the acceptance of your manuscript! Your careful revisions and thoughtful responses have further strengthened this valuable study. The additional analyses and clarifications enhance its scientific impact and clinical relevance. It has been a pleasure to review your work, and I look forward to its publication.

---

## [Editor Report · Acceptance letter]

PONE-D-24-39564R1

PLOS ONE

Dear Dr. Miyo,

I'm pleased to inform you that your manuscript has been deemed suitable for publication in PLOS ONE. Congratulations! Your manuscript is now being handed over to our production team.

Kind regards,

on behalf of

M.D., Ph.D. Tsutomu Kumamoto

Academic Editor

PLOS ONE